# Variation in detected adverse events using trigger tools: A systematic review and meta-analysis

Luisa C. Eggenschwiler[1☯], Anne W. S. Rutjes[2☯], Sarah N. Musy[1], Dietmar Ausserhofer[1,3], Natascha M. Nielen[1], René Schwendimann[1,4], Maria Unbeck[5,6], Michael Simon[1] *

1 Institute of Nursing Science (INS), Department Public Health (DPH), Faculty of Medicine, University of Basel, Basel, Switzerland, 2 Institute of Social and Preventive Medicine (ISPM), University of Bern, Bern, Switzerland, 3 College of Health Care-Professions Claudiana, Bozen-Bolzano, Italy, 4 Patient Safety Office, University Hospital Basel, Basel, Switzerland, 5 School of Health and Welfare, Dalarna University, Falun, Sweden, 6 Department of Clinical Sciences, Danderyd Hospital, Karolinska Institutet, Stockholm, Sweden

☯ These authors contributed equally to this work.
* m.simon@unibas.ch

## Abstract

### Background

Adverse event (AE) detection is a major patient safety priority. However, despite extensive research on AEs, reported incidence rates vary widely.

### Objective

This study aimed: (1) to synthesize available evidence on AE incidence in acute care inpatient settings using Trigger Tool methodology; and (2) to explore whether study characteristics and study quality explain variations in reported AE incidence.

### Design

Systematic review and meta-analysis.

### Methods

To identify relevant studies, we queried PubMed, EMBASE, CINAHL, Cochrane Library and three journals in the patient safety field (last update search 25.05.2022). Eligible publications fulfilled the following criteria: adult inpatient samples; acute care hospital settings; Trigger Tool methodology; focus on specialty of internal medicine, surgery or oncology; published in English, French, German, Italian or Spanish. Systematic reviews and studies addressing adverse drug events or exclusively deceased patients were excluded. Risk of bias was assessed using an adapted version of the Quality Assessment Tool for Diagnostic Accuracy Studies 2. Our main outcome of interest was AEs per 100 admissions. We assessed nine study characteristics plus study quality as potential sources of variation using random regression models. We received no funding and did not register this review.

**Data Availability Statement:** All data files are available from https://doi.org/10.5281/zenodo.4892518.

**Funding:** The author(s) received no specific funding for this work.

**Competing interests:** The authors have declared that no competing interests exist.

## Results

Screening 6,685 publications yielded 54 eligible studies covering 194,470 admissions. The cumulative AE incidence was 30.0 per 100 admissions (95% CI 23.9–37.5; $I^2$ = 99.7%) and between study heterogeneity was high with a prediction interval of 5.4–164.7. Overall studies' risk of bias and applicability-related concerns were rated as low. Eight out of nine methodological study characteristics did explain some variation of reported AE rates, such as patient age and type of hospital. Also, study quality did explain variation.

## Conclusion

Estimates of AE studies using trigger tool methodology vary while explaining variation is seriously hampered by the low standards of reporting such as the timeframe of AE detection. Specific reporting guidelines for studies using retrospective medical record review methodology are necessary to strengthen the current evidence base and to help explain between study variation.

## Introduction

For the last two decades, patient safety has become and remained a key issue for health care systems globally [1]. One major driver of patient harm in acute care hospitals are adverse events (AEs)—"unintended physical injury resulting from or contributed to by medical care that requires additional monitoring, treatment or hospitalization, or that results in death" [2]. Reported AE rates vary between 7% and 40% [3], increasing health care costs by roughly 10,000 Euros per index admission [4]. Considering that approximately 40% of admissions can be associated with AEs, it is likely that the consequences, both on health care service costs and on patient suffering, are underestimated [4, 5]. While some AEs are hardly avoidable, others are: studies have indicated that 6%–83% of AEs are deemed to be preventable [6, 7].

Retrospective medical record reviews are commonly used when collecting data about patient safety such as AEs. Medical record review methodology using available data [8], was found to identify more AEs when compared with other methods [9, 10], can be repeated over time and can target specific AE types, or the overall AE rate [11].

There are several medical record review methods, and the most used ones are the Harvard Medical Practice Study (HMPS) methodology [12], with subsequently modifications [13], and the Global Trigger Tool (GTT) [2]. The GTT, popularised by the Institute for Healthcare Improvement (IHI) in the US, was primarily designed as a measurement tool in clinical practice to estimate and track AE rates over time, extending beyond traditional incident reports, and aiming to measure the effect of safety interventions [14, 15]. The GTT includes a two-step medical record review process. In the first step, knowledgeable hospital staff—mainly nurses, conduct primary reviews to identify potential AEs using predefined triggers as outlined in the GTT guidance. In the second step, physicians verify the reviews from the first step and authenticate their consensus. A "trigger" (or clue) is either a specific term or an event in a medical record that could indicate the occurrence of an AE, e.g., readmissions within 30 days or pressure ulcers [2]. Its main methodological advantage is that it is an open, inductive process, sensitive to detect various types of AEs [2]. GTT based studies typically report inter-rater reliability coefficients that represent satisfactory reliability (kappa 0.34 to 0.89; mean: 0.65) [16].

GTT's triggers are grouped into six modules (e.g., Care Module, Medication Module). Some researchers use all six of these [17, 18] while most use only those relevant to their setting [19, 20]. Yet others either create additional modules (e.g., Oncology Module [21, 22]) or develop modified versions tailored specifically to their patient and care settings [3, 23]. While former versions diverge too importantly from the original GTT to label it as GTT, they are still considered as trigger tools (TTs).

When using the GTT outside of the USA, even in cases where translation is unnecessary, triggers need to be adapted to reflect local norms (e.g., blood level limits). Additionally, medication labels need to be adjusted as appropriate [24, 25]. Although the GTT was developed as a manual method, with the rise of electronic health records, the GTT process can be semi or fully automated [26].

Recent systematic reviews focussing on AEs detected via GTT or TT showed high detection rate variability [3, 6, 26]. Some of this variability may reflect differences in the studies' methodological features. Adaptations in triggers, review processes or patient record selection protocols might influence detection rates, thereby impacting the comparability of detected AEs. Such differences in medical record review methodology have not yet been systematically addressed. Therefore, this study has two aims: (1) to synthesize the evidence identified by the TT methodology regarding AE incidence in acute care inpatient settings; and (2) to explore whether between study variation in the incidence of AEs can be explained by study characteristics and study quality.

## Methods

### Design

This systematic review and meta-analyses adhered to the preferred reporting items for PRISMA guideline [27, 28].

### Search strategy and information sources

Our search strategy was developed and validated using methods suggested by Hausner et al. [29, 30]. This involves generating a test set, developing and validating a search strategy and documenting the strategy using a standardized approach [30]. The medical subject headings (MeSH) and keywords for titles and abstracts in our search string were: *(trigger[tiab] OR triggers[tiab]) AND (chart[tiab] OR charts[tiab] OR identif\* [tiab] OR record[tiab] OR records [tiab]) AND (adverse[tiab] OR medical error[mh])*. We used this to query four electronic databases: PubMed, EMBASE, CINAHL and Cochrane Library. In addition, we also hand-searched the top three journals publishing about GTT/TT (BMJ Quality & Safety; Journal of Patient Safety; International Journal for Quality in Health) and screened all authors' personal libraries. In all searches, publication dates were unrestricted. The detailed search strategy used for this review and further explanations on chosen journals is published in Musy et al. [26]. The index search was conducted in November 2015, additional five update searches in April 2016, July 2017, January 2020, September 2020, and the latest update on May 25 2022.

### Eligibility criteria

We included publications fulfilling six criteria:1. publication in English, French, German, Italian or Spanish; 2. adult inpatient samples; 3. acute care hospital settings; 4. medical record review performed manually via GTT or other TT methods; 5. specialties in internal medicine, surgery (including orthopaedics), oncology, or any combination of these (mixed); and 6.

outcome data relevant to our study, e.g., number of detected AEs. Systematic reviews and studies addressing only *adverse drug events* or exclusively deceased patients were excluded.

## Study selection and data extraction

Titles and abstracts were screened independently by two researchers in a first round if they included any information on GTT or TT and in a second round on the eligibility criteria. After screening the titles and abstracts, two researchers individually assessed the full-text articles for eligibility. To ensure high-quality data entry, data were extracted by one researcher and verified by a second. Information on study characteristics (e.g., number of admissions, setting, patient demographics) and patient outcomes (incidence, preventability) were collected into an online data collection instrument (airtable.com). Where studies of authors of this report were considered, a pair without direct involvement in the primary study was chosen to abstract and appraise the study. Differences between researchers were then discussed in the research group to reach consensus.

Our main outcome of interest was AEs per 100 admissions ((number of AEs / number of admissions) * 100). In addition, we included three secondary outcomes: AEs per 1,000 inpatient days ((number of AEs / number of inpatient days) * 1,000), the percentage of admissions with one or more AEs (number of admissions with ≥1 AE / number of admissions) and percentage of preventable AEs (number of preventable AEs / number of AEs). We included nine TT methodology characteristics in our statistical analysis to assess their potentially influence on AE detection rates. We categorized these under four headings: setting (type of hospital, type of specialty), patient characteristics (age, length of stay), design (AE definition, timeframe of AE detection, commission/ omission) and reviewer (training, experience). Definitions of our variables, our categorisations of the selected characteristics and our rationale for the chosen variable and its categorisation are available in Table 1.

## Quality assessment

To assess the risk of bias and applicability-related concerns for each included study, we developed and piloted a quality assessment tool (QAT) (see S1 File). This was inspired by the Quality Assessment Tool for Diagnostic Accuracy Studies 2 (QUADAS-2) tool and by the QAT developed by Musy et al. [41]. While assessing our included studies, we used both QUADAS-2 tool dimensions: the risk of bias and applicability-related concerns [41]. We assessed five domains: 1) patient selection; 2) rater or reviewer; 3) trigger tool method; 4) outcomes; and 5) flow and timing. Following the QUADAS-2 structure each domain included standardised signalling questions to help researchers' rate each of the two dimensions, i.e., risk of bias and applicability-related concerns. Possible dimension classifications were low, high, or unclear. For each study, a QAT was completed by one researcher and reviewed by a second. To reach consensus, differences were discussed between the two and, if necessary, within the research group.

## Statistical analysis

To analyse and plot our results we used R version 4.1.3 on Linux [42] with the meta [43] and metafor [44] packages. We determined the number of AEs per 100 admissions and the number of AEs per 1,000 patient days from the reported data. If the number of AEs was not explicitly described, we calculated it from the reported estimate of AEs per 100 admissions and number of patient admissions. The number of patient days could for example be calculated from the *total number of AEs per 1,000 patient days*. For studies published by this study's co-authors or in some cases by their research colleagues, when samples overlapped, we asked them for additional information in order to avoid double counting of admissions and AEs [34, 45, 46]. Pooled estimates for AEs per 100 admissions and AEs per 1,000 patient days were derived

**Table 1. Study characteristics for stratified analysis.**

| Variable | Definition | Categorisation | Rationale |
|---|---|---|---|
| Setting | | | |
| Hospital | Type of hospital | Academic hospital | We reasoned that academic hospitals tend to receive more severely ill or complex patients at higher risk of experiencing AEs when compared to other hospital types [31]. |
| | | Non-academic hospital | |
| | | Mixed | |
| | | Not reported | |
| Specialty | Type of unit | Internal medicine | We expected the AE incidence to vary by type of specialty. We combined surgical and orthopaedical units as an important fraction of admitted orthopaedical patients was expected to undergo surgical interventions. Mixed = a combination of the three categories mentioned above or combined with other specialties [3, 32, 33]. |
| | | Surgery and orthopaedics | |
| | | Oncology | |
| | | Mixed | |
| | | Not reported | |
| Patient characteristics | | | |
| Age | Mean or median age of patients at admission | > 70 years | Multi-morbidity and polypharmacy are expected to occur more often in elderly patients. We anticipated patients with multimorbid conditions or polypharmacy to be at higher risk for AEs [31, 33, 34]. |
| | | ≤ 70 years | |
| | | Not reported | |
| Length of stay (LOS) | Mean or median length of hospital stay | LOS > 5 days | Patients with longer LOS are at higher risk of experiencing AEs. As the average LOS in the US and many European countries ranges between 4 and 6 days, we chose a cut-off at five days [23, 35, 36]. |
| | | LOS ≤ 5 days | |
| | | Not reported | |
| Design | | | |
| AE definition | IHI AE definition | IHI like | We expected that differences in the AE definition between studies lead to variation in estimates of AE incidence [33, 37]. Definition: "unintended physical injury resulting from or contributed to by medical care that requires additional monitoring, treatment or hospitalisation, or that results in death" [2] |
| | | "Narrower" than IHI GTT | |
| | | "Wider" than IHI GTT | |
| | | Not reported | |
| Timeframe of AE detection | Definition of the time period in which AEs were detected. | Hospital stay plus time after discharge | The frequency of AEs varies depending on the timeframe and setting considered, i.e., before and after index admission [38]. If a study reported AEs only during hospitalisation, it was categorized into the category "hospital stay plus time before admission". |
| | | Hospital stay plus time before admission | |
| | | Hospital stay plus time before and after admission | |
| | | Not reported | |
| Commission and omission | Evaluation of commission or omission of care | Inclusion of commission only | The IHI GTT focuses on AEs related to commission (doing the wrong thing), however in recent years authors have included omissions (failing to do the right thing). Including omissions in medical record reviews may lead to more AEs detected [3]. |
| | | Inclusion of commission and omission | |
| | | Not reported | |
| Reviewer | | | |
| Training | The reviewer's training before starting with data collection | Training plus pilot phase | We reasoned that trained and/or experienced reviewers were less likely to miss AEs than untrained or unexperienced reviewers [37, 39, 40]. |
| | | Training only | |
| | | No training | |
| | | Not reported | |
| Experience | The reviewer's experience in application of the GTT method or similar medical record review method. | GTT or medical record review experience | |
| | | No experience | |
| | | Not reported | |

AE, Adverse event; GTT, Global Trigger Tool; IHI, Institute for Healthcare Improvement; LOS, length of stay

using a random effects Poisson regression approach within the R *metarate* function [43, 44]. With the R *metaprop* function, a random effects logistic regression model was used to obtain summary estimates and confidence intervals (derived by the Wilson method) for the outcomes expressed as percentage of admissions with ≥1 AE and percentage of preventable AEs [43].

**Subgroup analysis.** Heterogeneity was explored by stratified analyses, which were performed on the main outcome measure, i.e. number of AEs per 100 admissions to evaluate the influence of the nine study characteristics: type of hospital, type of specialty, patient age, length of stay, AE definition, timeframe of AE detection, commission and omission, reviewer training, and reviewer experience. In addition, we analysed five elements relating to risk of bias and the three for applicability-related concerns. P-values were derived from the likelihood ratio test for model fit (p < 0.05 was considered significant). Furthermore, between study heterogeneity was evaluated visually and by calculating the prediction intervals [47, 48]. To assess the risk of publication bias related to small study size, we created a funnel plot regressing the logit of the AEs per 100 admissions on the standard error, assessed the symmetry of the distribution and performed the Egger test [49].

## Results

The index search and update searches produced 9,780 returns. Deleting duplicates left 6,685 separate entries. The more detailed screening process left 54 studies, which were published in 72 publications [5, 9, 10, 14, 15, 17–22, 24, 34, 37–40, 45, 46, 50–102]. Fig 1 depicts the complete review procedure.

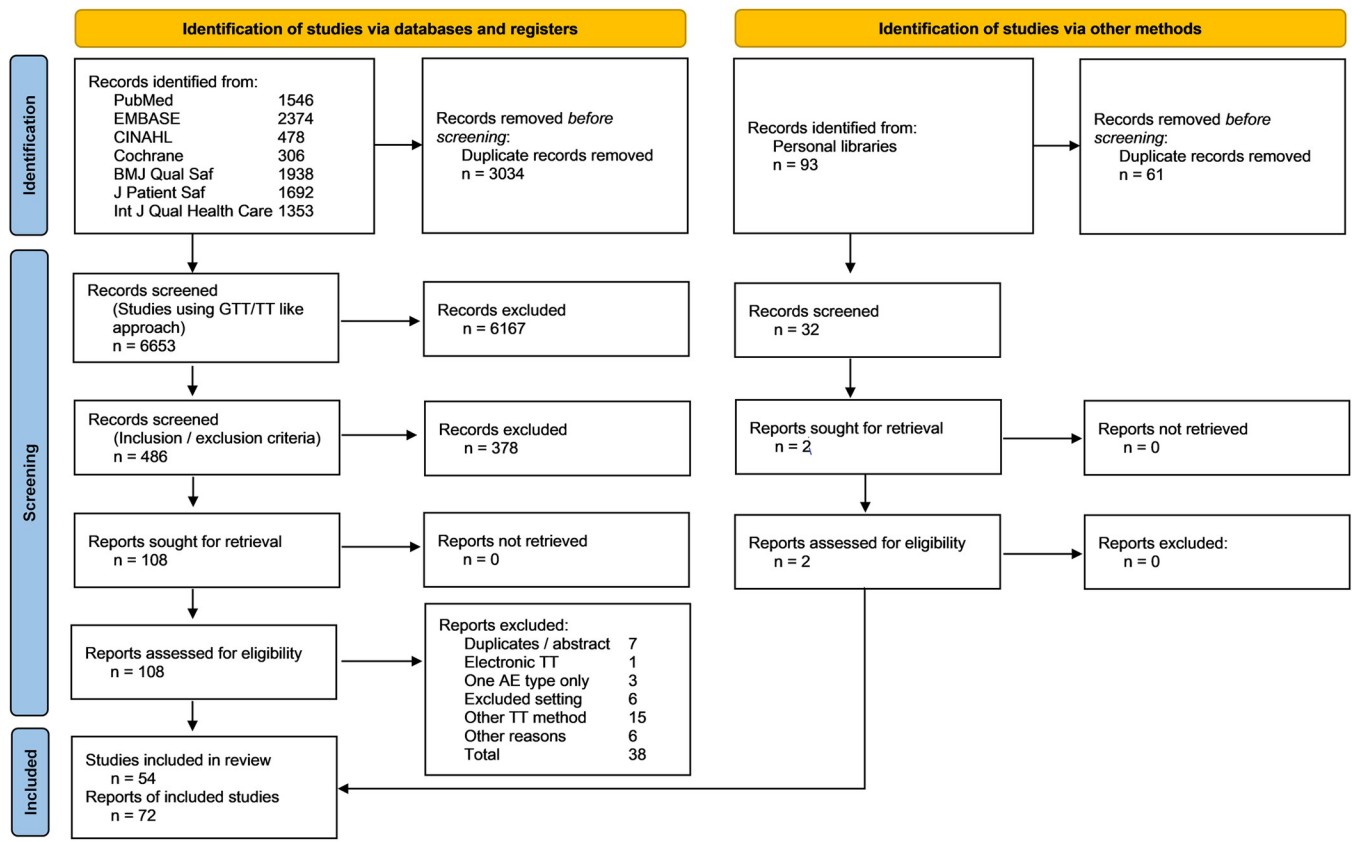

**Fig 1. Flow diagram of literature search and included studies.** From [27] (GTT, Global Trigger Tool, TT, Trigger Tool).

## Study characteristics

The 54 included studies were all published between 2009 and 2022. Their study periods ranged from one month to six years (Table 2). They were conducted in 26 countries, most of them in Europe (34 studies, 63%), followed by the US (12 studies, 22%) and Others (8 studies, 15%).

Four studies (7%) did not report their clinical specialties [10, 17, 71, 77]. For those remaining, almost half (24 studies, 44%) involved mixed specialties. One study included no information on the number of included records [40]. The numbers of included records ranged from 50 to 56,447. Overall, we included 194,470 index admissions in our report.

Table 3 illustrates AE rates' key characteristics. In seven studies, we could not retrieve the main outcome measure AEs per 100 admissions [14, 24, 40, 55, 70, 80, 94]; for the remaining 47, rates ranged from 2.5 to 140 per 100 admissions. Per 1,000 patient days, the 36 (67%) studies with sufficient data yielded counts ranging from 12.4 to 139.6. And in the 48 studies whose data allowed us to calculate percentages of admissions with one or more AEs, these ranged from 7% to 69%. AE preventability percentages, which 37 studies (69%) reported, ranged from 7% to 93%; however, four of these studies provided no relevant raw data [21, 45, 55, 56].

## Quality assessment

Our quality assessment results (Fig 2) indicate that most of the domains of the risk of bias are rated as *low* (range: 48%–93%). However, the "patient selection" and "reviewer" domains received respectively 15% and 13% *high* ratings—considerably more than the other domains (range: 2%–6%). In two domains, risk of bias was largely *unclear*: "reviewer" and "trigger tool method" received this rating respectively in 39% and 30% of cases.

Overall applicability-related concerns were predominantly *low* (range of domains: 65%–87%). *High* ratings were most prevalent (17%) in the "patient selection" domain; *unclear* ratings were most common (28%) for "reviewer". Quality assessment results on study-level are provided in S1 Table.

## Summary estimates from meta-analyses

The forest plot in Fig 3 presents AEs per 100 admissions by sample size. Forty-five samples from single countries contributed, as well as two multi-country (n = 10) samples [61, 71]. The summary estimate was 30.0 AEs per 100 admissions (95% CI 23.9–37.5). Visual inspection of the forest plot indicated a high level of between study heterogeneity, which was confirmed by an $I^2$ of 99.7% (95% CI 99.7–99.7). The prediction interval ranged from 5.4 to 164.7 AEs per 100 admissions. Four studies had exceptionally high detection rates [19, 20, 38, 87]. At the opposite side, seven study samples reported fewer than ten AEs per 100 admissions [17, 56, 71].

S1–S3 Figs present additional forest plots for the three secondary outcomes, respectively AEs per 1,000 patient days (n = 36 studies), percentages of admissions with AEs (n = 48 studies), and percentages of preventable AEs (n = 33 studies). Our meta-analysis showed a summary estimate of 48.3 AEs per 1,000 patient days (95% CI 40.4–57.8) with high level of between study heterogeneity (prediction interval 15.9–147.0). Twenty-six percent of admissions experienced one or more AEs (95% CI 22.0–29.5, prediction interval 7.8–58.3). Within the studies that rated preventability, 62.6% of AEs were classified as preventable (95% CI 54.0–70.5, prediction interval 16.8–93.3). Similarly, visual inspection indicated a high between study heterogeneity. Funnel plot exploration did not suggest evidence for publication bias or other biases related to small study size (P from Egger test = 0.3, S4 Fig).

**Effect of study characteristics.** Eight of nine analysed study characteristics explained part of the heterogeneity between studies (Fig 4).

**Table 2. Characteristics of the 54 included studies.** Sorted by continent; within continent alphabetically by country code, and within the country by year.

| Study | Country | Study period number of months | Sample size number of records | Patient age | Length of stay | Clinical specialty | Type of hospital | Timeframe of AE detection |
|---|---|---|---|---|---|---|---|---|
| **Europe** | | | | | | | | |
| Hoffmann 2018 [86] | AUT | 12 | 239 | ≤70 years | > 5 days | SURG | Academic | NR |
| Grossmann 2019 [19] | CHE | 12 | 240 | ≤70 years | > 5 days | MED | Academic | Stay + Before |
| Gerber 2020 [21] | CHE | 1.5 | 224 | ≤70 years | ≤ 5 days | ONCO | Mixed | Stay + After + Before |
| Nowak 2022 [100] | CHE | 12 | 150 | >70 years | > 5 days | MED | Academic | Stay + After + Before |
| Lipczak 2011 [69, 88] | DNK | 6 | 572 | NR | NR | ONCO | NR | NR |
| von Plessen 2012 [40] | DNK | 18 | NR | ≤70 years | NR | MIX | NR | NR |
| Mattson 2014 [22, 68] | DNK | 12 | 240 | NR | NR | ONCO | Academic | NR |
| Bjorn 2017 [52] | DNK | 6 | 120 | NR | NR | MIX | Academic | NR |
| Brösterhaus 2020 [82] | DEU | 2 | 80 | NR | > 5 days | SURG | Academic | NR |
| Suarez 2014 [63, 91] | ESP | 72 | 1,440 | NR | NR | MIX | Non-aca | NR |
| Guzman Ruiz 2015 [64, 67] | ESP | 12 | 291 | >70 years | > 5 days | MED | Non-aca | NR |
| Perez Zapata 2015 [53, 66] | ESP | 12 | 350 | ≤70 years | NR | SURG | Academic | NR |
| Toribio-Vicente 2018 [94] | ESP | 12 | 233 | NR | NR | MIX | Academic | NR |
| Kaibel 2020 [97] | ESP | 12 | 251 | ≤70 years | ≤ 5 days | SURG | Academic | Stay + After |
| Menendez-Fraga 2021 [98] | ESP | 12 | 240 | >70 years | > 5 days | MED | Academic | Stay + After |
| Perez Zapata 2022 [101] | ESP | 9 | 1132 | ≤70 years | > 5 days | SURG | Mixed | Stay + After |
| Mayor 2017 [56] | GBR | 36 | 4,833 | ≤70 years | NR | MIX | Mixed | NR |
| Mortaro 2017 [60] | ITA | 66 | 513 | ≤70 years | NR | MIX | Non-acad | NR |
| Cihangir 2013 [70] | NLD | 12 | 129 | NR | NR | ONCO | NR | NR |
| Deilkas 2015 [24, 81, 92] | NOR | 34 | 29,865 | NR | NR | MIX | Mixed | NR |
| Farup 2015 [80] | NOR | 24 | 272 | ≤70 years | > 5 days | MED | Non-acad | NR |
| Mevik 2016 [57, 58] | NOR | 12 | 1,680 | ≤70 years | > 5 days | MIX | Academic | Stay + After + Before |
| Haukland 2017 [54, 85] | NOR | 48 | 812 | ≤70 years | > 5 days | ONCO | Non-acad | NR |
| Deilkas 2017 [61] | NOR | 12 | 10,986 | NR | NR | MIX | Mixed | NR |
| Pierdevara 2020 [102] | PRT | 9 | 176 | >70 years | > 5 days | MIX | Mixed | NR |
| Schildmeijer 2012 [72] | SWE | 8 | 50 | ≤70 years | ≤ 5 days | MIX | NR | NR |
| Unbeck 2013 [37] | SWE | 12 | 350 | ≤70 years | ≤ 5 days | SURG | Academic | Stay + After + Before |
| Rutberg 2014 [73] | SWE | 48 | 960 | ≤70 years | > 5 days | MIX | Academic | Stay + After + Before |
| Nilsson 2016 [46] | SWE | 12 | 3,301 | ≤70 years | > 5 days | SURG | Mixed | NR |
| Rutberg 2016 [34] | SWE | 24 | 4,994 | >70 years | > 5 days | SURG | Mixed | Stay + After + Before |
| Deilkas 2017 [61] | SWE | 12 | 19,141 | NR | NR | MIX | Mixed | NR |
| Nilsson 2018 [45, 84] | SWE | 48 | 56,447 | ≤70 years | > 5 days | MIX | Mixed | NR |
| Hommel 2020 [20, 89, 90] | SWE | 36 | 1,998 | >70 years | > 5 days | SURG | Mixed | Stay + After |
| Kelly-Pettersson 2020 [96] | SWE | 24 | 163 | >70 years | > 5 days | SURG | Academic | Stay + After |
| Kurutkan 2015 [18] | TUR | 12 | 229 | ≤70 years | ≤ 5 days | MIX | Academic | NR |
| **North America** | | | | | | | | |
| Griffin 2008 [83] | USA | 12 | 854 | NR | NR | SURG | NR | NR |
| Naessens 2010 [9, 14] | USA | 25 | 1,138 | NR | NR | MIX | Academic | NR |

*(Continued)*

**Table 2.** (Continued)

| Study | Country | Study period number of months | Sample size number of records | Patient age | Length of stay | Clinical specialty | Type of hospital | Timeframe of AE detection |
|---|---|---|---|---|---|---|---|---|
| Landrigan 2010 [39, 77] | USA | 72 | 2,341 | ≤70 years | NR | NR | Mixed | NR |
| Classen 2011 [10] | USA | 1 | 795 | ≤70 years | ≤ 5 days | NR | Mixed | NR |
| Garrett 2013 [5, 79] | USA | 36 | 17,295 | ≤70 years | ≤ 5 days | MIX | Mixed | NR |
| O'Leary 2013 [74] | USA | 12 | 250 | ≤70 years | > 5 days | MED | Academic | NR |
| Kennerly 2014 [15, 50, 78] | USA | 60 | 9,017 | NR | NR | MIX | Non-acad | Stay + After + Before |
| Mull 2015 [76] | USA | 4 | 273 | ≤70 years | > 5 days | MIX | Non-acad | NR |
| Croft 2016 [38, 59] | USA | 11 | 296 | ≤70 years | ≤ 5 days | MIX | Academic | Stay + After + Before |
| Lipitz-Snyderman 2017 [55] | USA | 12 | 400 | ≤70 years | NR | ONCO | Academic | NR |
| Zadvinskis 2018 [95] | USA | 1 | 317 | ≤70 years | ≤ 5 days | MIX | Academic | NR |
| Sekijima 2020 [93] | USA | 4 | 300 | ≤70 years | > 5 days | MED | Academic | NR |
| **Other** | | | | | | | | |
| Moraes 2021 [99] | BRA | 1 | 220 | ≤70 years | > 5 days | MIX | Academic | Stay + After |
| Xu 2020 [62] | CHN | 12 | 240 | ≤70 years | > 5 days | MIX | Academic | Stay + After |
| Hu 2019 [87] | CHN | 12 | 480 | >70 years | > 5 days | MIX | Academic | NR |
| Wilson 2012 [71]* | EGY | 12 | 1,358* | ≤70 years | NR | NR | Mixed | NR |
| | JOR | | 3,769 | | | | | |
| | KEN | | 1,938 | | | | | |
| | MAR | | 984 | | | | | |
| | ZAF | | 931 | | | | | |
| | SDN | | 3,977 | | | | | |
| | RUN | | 930 | | | | | |
| | YEM | | 1,661 | | | | | |
| Najjar 2013 [75] | ISR | 4 | 640 | ≤70 years | ≤ 5 days | MIX | Mixed | NR |
| Hwang 2014 [17] | KOR | 6 | 629 | ≤70 years | > 5 days | NR | Academic | NR |
| Asavaroengchai 2009 [51] | THA | 1 | 576 | ≤70 years | ≤ 5 days | MIX | Academic | NR |
| Müller 2016 [65] | ZAF | 8 | 160 | ≤70 years | > 5 days | MED | Academic | Stay + Before |

NR, not reported; MED, internal medicine; MIX, mixed; ONCO, oncology; SURG, surgery/orthopaedics; Academic, academic hospital; Non-acad, non-academic hospital; Stay + After, hospital stay plus time after discharge; Stay + Before, hospital stay plus time before admission; Stay + After + Before, hospital stay plus time before and after admission; *After coding these countries A-H, this studies' authors linked each number directly to a letter, but failed to link each letter to a particular country, therefore it is impossible to reconcile these numbers with the countries listed.

As for the type of hospital study characteristic, *academic medical centres* (n = 25, 45%) detected more AEs per 100 admissions than *non-academic hospitals* (respectively 47.1, 95% CI 36.6–60.5 and n = 6, 11%; 35.8, 95% CI 30.8–41.7), but as the summary estimate for mixed types of hospitals (n = 21, 38%; 17.0, 95% CI 11.7–24.8) is lower than either academic and non-academic hospitals, this association is likely confounded by a third feature. For type of clinical specialty, the significant differences within categories were driven by the *not reported* category (n = 11, 20%), which had fewer AEs per 100 admissions compared to the others (10.6, 95% CI 6.8–16.7). The *internal medicine* specialty (n = 7, 13%) had the highest number of AEs per 100 admissions (56.4, 95% CI 40.5–78.5), followed by *surgery/orthopaedics* (n = 11, 20%; 41.7, 95% CI 29.5–59.0). O*ncology* (n = 4, 7%) had numbers similar to those of the *mixed* designation (respectively 40.0, 95% CI 26.2–61.3 vs. 33.5, 95% CI 25.0–44.8).

**Table 3. Main characteristics of adverse events (AE) rates.**

| Study | AEs per 100 admissions | AEs per 1,000 patient days | % of admissions with $\geq$ 1 AE | % of preventable AEs out of all AEs |
|---|---|---|---|---|
| Wilson 2012 [71], Country B | 2.5 | NR | NR | 83.9 |
| Wilson 2012 [71], Country F | 5.5 | NR | NR | 84.4 |
| Wilson 2012 [71], Country A | 6.0 | NR | NR | 72.8 |
| Hwang, 2014 [17] | 7.8 | 12.4 | 7.2 | 61.2 |
| Wilson 2012 [71], Country E | 8.2 | NR | NR | 55.3 |
| Wilson 2012 [71], Country G | 8.3 | NR | NR | 85.7 |
| Mayor, 2017 [56] | 8.9 | NR | 8.0 | AEs detected by TT not reported separately |
| Najjar, 2013 [75] | 14.2 | NR | 14.2 | 59.3 |
| Nilsson, 2018 [45, 84]$ | 14.4 | 20.2 | 11.4 | Included sample not reported separately |
| Wilson 2012 [71], Country C | 14.5 | NR | NR | 76.9 |
| Wilson 2012 [71], Country D | 14.8 | NR | NR | 85.6 |
| Deilkas, 2017 [61] (NOR) | 15.2 | NR | 13.0 | NR |
| Griffin, 2008 [83] | 16.2 | NR | 14.6 | NR |
| Deilkas, 2017 [61] (SWE) | 16.8 | NR | 14.4 | NR |
| Wilson 2012 [71], Country H | 18.4 | NR | NR | 93.1 |
| Rutberg, 2016 [34]$ | 19.0 | 27.0 | 14.7 | 73.4 |
| Nilsson, 2016 [46]$ | 19.9 | 29.6 | 15.4 | 62.5 |
| Zadvinskis, 2018 [95]‡ | 21.1 | 68.9 | NR | NR |
| Mattson, 2014 [22, 68] | 23.3 | 37.4 | 20.8 | NR |
| Landrigan, 2010 [39, 77] | 25.1 | 56.5 | 18.1 | 61.9 |
| Mevik, 2016 [57, 58] | 26.6 | 39.3 | 20.7 | NR |
| Rutberg, 2014 [73]$ | 28.2 | 33.2 | 20.5 | 71.2 |
| Xu, 2020 [62] | 29.2 | 32.1 | 22.5 | NR |
| Kurutkan, 2015 [18] | 29.3 | 80.72 | 17.0 | 64.2 |
| Suarez, 2014 [63, 91] | 29.4 | 24.5 | 23.3 | 65.8 |
| Schildmeijer, 2012 [72] | 30.0 | 45.3 | 20.0 | 60.0 |
| Mortaro, 2017 [60]* | 30.4 | 31.9 | 21.6 | NR |
| Haukland, 2017 [54, 85] | 31.2 | 37.1 | 24.3 | NR |
| O'Leary, 2013 [74] | 34.4 | NR | 21.6 | 7.0 |
| Brösterhaus, 2020 [82]* | 36.2 | 31.6 | 27.5 | NR |
| Müller, 2016 [65] | 36.9 | 25.8 | 24.4 | 47.5 |
| Garrett 2013 [5, 79]‡ | 38.0 | 85.0 | 26.0 | NR |
| Kennerly 2014 [15, 50, 78] | 38.0 | 61.3 | 32.1 | 18.0 |
| Unbeck, 2013 [37]$ | 39.1 | 74.1 | 28.0 | 80.3 |
| Mull, 2015 [76] | 39.9 | 52.4 | 21.6 | NR |
| Asavaroengchai, 2009 [51] | 41.0 | 52.9 | 24.0 | 55.9 |
| Classen, 2011 [10] | 44.5 | NR | NR | NR |
| Lipczak, 2011 [69, 88] | 45.5 | NR | NR | NR |
| Perez Zapata, 2015 [53, 66] | 46.0 | NR | 31.7 | 54.7 |
| Sekijima, 2020 [93]* | 46.3 | 73.7 | 28.3 | NR |
| Guzman Ruiz, 2015 [64, 67] | 51.2 | 63.0 | 35.4 | 32.2 |
| Perez Zapata, 2022 [101] | 52.9 | NR | 31.5 | 34 |
| Menendez-Fraga, 2021 [98] | 57.1 | 49.8 | 44.6 | 49.6 |
| Hoffmann, 2018 [86]* | 61.9 | 31.5 | 33.5 | NR |
| Kelly-Pettersson, 2020 [96]$ | 62.6 | 104.2 | 38.0 | 60.8 |
| Nowak, 2022 [100] | 72.0 | 90.6 | 42.7 | 54.6 |
| Gerber, 2020 [21] | 75.4 | 106.6 | 42.0 | Included sample not reported separately |

*(Continued)*

**Table 3.** (Continued)

| Study | AEs per 100 admissions | AEs per 1,000 patient days | % of admissions with ≥ 1 AE | % of preventable AEs out of all AEs |
|---|---|---|---|---|
| Kaibel, 2020 [97] | 76.1 | NR | 45.8 | 92.1 |
| Pierdevara, 2020 [102] | 80.7 | 42.1 | NR | NR |
| Bjorn, 2017 [52]• | 81.7 | 139.6 | 44.2 | NR |
| Moraes, 2021 [99] | 90.5 | 76.1 | 40.9 | NR |
| Hommel, 2020 [20, 89, 90]$ | 105.9 | 93.2 | 58.6 | 75.9 |
| Croft, 2016 [38, 59] | 114.2 | NR | NR | 50.0 |
| Hu, 2019 [87] | 127 | 22.4 | 68.5 | 50.8 |
| Grossmann, 2019 [19] | 140 | 95.7 | 60.0 | 29.2 |
| Cihangir, 2013 [70]* | NR | NR | 36.4 | NR |
| Deilkas, 2015 [24, 81, 92]* | NR | NR | 15.1 | NR |
| Farup, 2015 [80]* | NR | NR | 14.0 | NR |
| Lipitz-Snyderman, 2017 [55] | NR | NR | 36.0 | AEs detected by TT not reported separately |
| Naessens, 2010 [9, 14] | NR | NR | 27.0 | NR |
| Toribio-Vicente, 2018 [94]* | NR | NR | 20.2 | NR |
| von Plessen, 2012 [40] | NR | 59.8 | 25# | NR |

NR, not reported; TT, Trigger Tool.

* Pooled estimate.

• Mean estimate.

‡ Calculated total number of AEs.

$ Additional outcome data included.

# Original data reported.

Older patients *(mean > 70 years;* n = 8, 15%*)* had a higher incidence of AEs than younger ones *(mean ≤ 70 years;* n = 38, 69%*)*, although only eight studies specifically investigated older patients (respectively 63.7, 95% CI 43.6–93.0 and 25.9, 95% CI 19.6–34.2). As occurred with

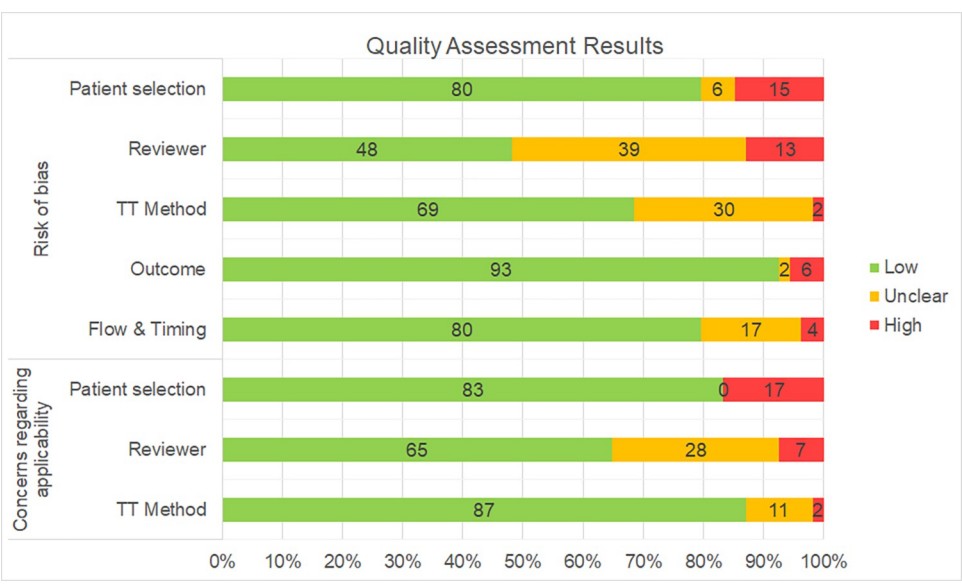

**Fig 2. Quality assessment of all included studies.** Assessments are presented in *risk of bias* and *applicability-related concerns*. (TT method, Trigger Tool method).

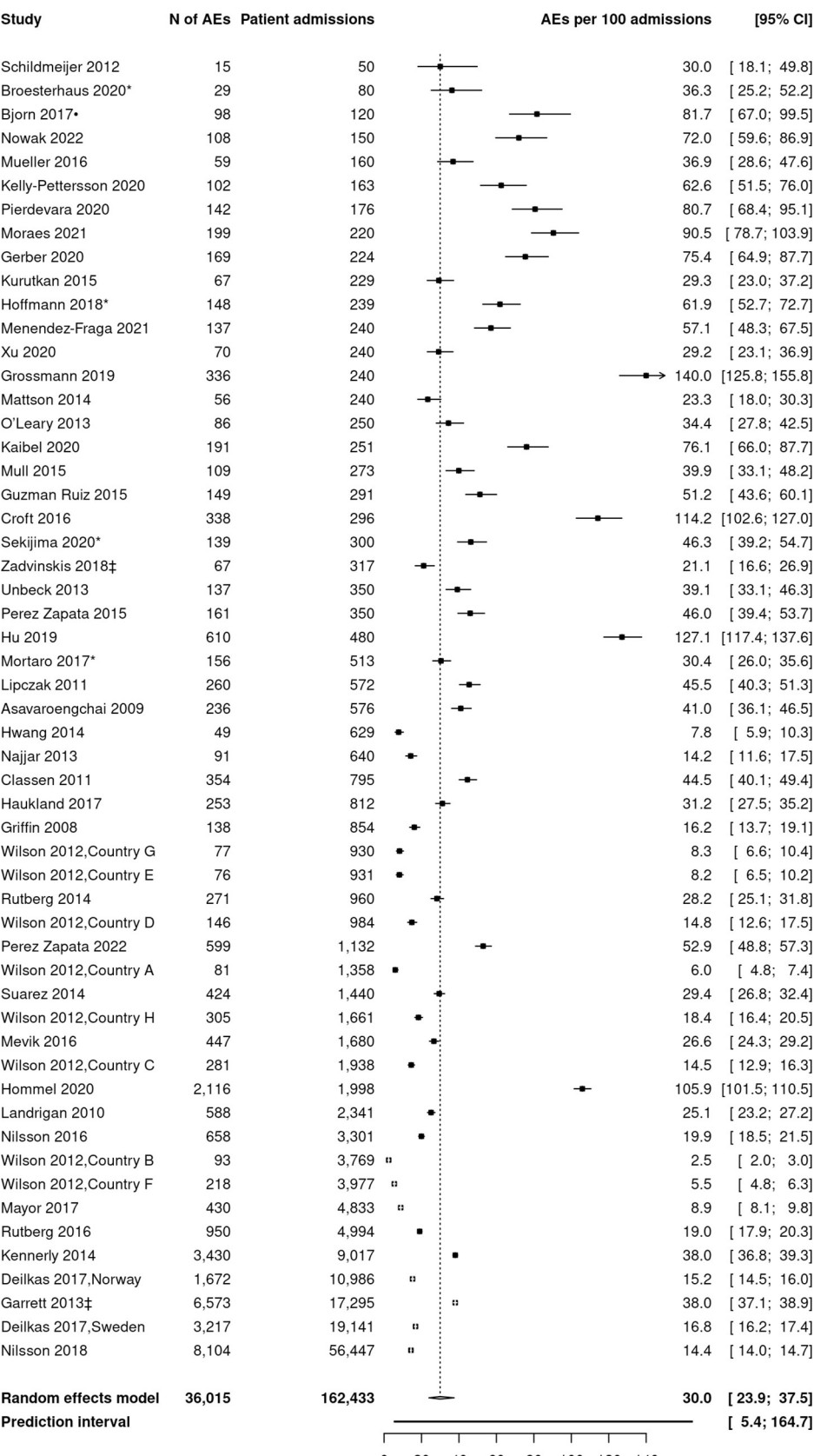

| Study | N of AEs | Patient admissions | AEs per 100 admissions | [95% CI] |
|---|---|---|---|---|
| Schildmeijer 2012 | 15 | 50 | 30.0 | [ 18.1; 49.8] |
| Broesterhaus 2020* | 29 | 80 | 36.3 | [ 25.2; 52.2] |
| Bjorn 2017• | 98 | 120 | 81.7 | [ 67.0; 99.5] |
| Nowak 2022 | 108 | 150 | 72.0 | [ 59.6; 86.9] |
| Mueller 2016 | 59 | 160 | 36.9 | [ 28.6; 47.6] |
| Kelly-Pettersson 2020 | 102 | 163 | 62.6 | [ 51.5; 76.0] |
| Pierdevara 2020 | 142 | 176 | 80.7 | [ 68.4; 95.1] |
| Moraes 2021 | 199 | 220 | 90.5 | [ 78.7; 103.9] |
| Gerber 2020 | 169 | 224 | 75.4 | [ 64.9; 87.7] |
| Kurutkan 2015 | 67 | 229 | 29.3 | [ 23.0; 37.2] |
| Hoffmann 2018* | 148 | 239 | 61.9 | [ 52.7; 72.7] |
| Menendez-Fraga 2021 | 137 | 240 | 57.1 | [ 48.3; 67.5] |
| Xu 2020 | 70 | 240 | 29.2 | [ 23.1; 36.9] |
| Grossmann 2019 | 336 | 240 | 140.0 | [125.8; 155.8] |
| Mattson 2014 | 56 | 240 | 23.3 | [ 18.0; 30.3] |
| O'Leary 2013 | 86 | 250 | 34.4 | [ 27.8; 42.5] |
| Kaibel 2020 | 191 | 251 | 76.1 | [ 66.0; 87.7] |
| Mull 2015 | 109 | 273 | 39.9 | [ 33.1; 48.2] |
| Guzman Ruiz 2015 | 149 | 291 | 51.2 | [ 43.6; 60.1] |
| Croft 2016 | 338 | 296 | 114.2 | [102.6; 127.0] |
| Sekijima 2020* | 139 | 300 | 46.3 | [ 39.2; 54.7] |
| Zadvinskis 2018‡ | 67 | 317 | 21.1 | [ 16.6; 26.9] |
| Unbeck 2013 | 137 | 350 | 39.1 | [ 33.1; 46.3] |
| Perez Zapata 2015 | 161 | 350 | 46.0 | [ 39.4; 53.7] |
| Hu 2019 | 610 | 480 | 127.1 | [117.4; 137.6] |
| Mortaro 2017* | 156 | 513 | 30.4 | [ 26.0; 35.6] |
| Lipczak 2011 | 260 | 572 | 45.5 | [ 40.3; 51.3] |
| Asavaroengchai 2009 | 236 | 576 | 41.0 | [ 36.1; 46.5] |
| Hwang 2014 | 49 | 629 | 7.8 | [ 5.9; 10.3] |
| Najjar 2013 | 91 | 640 | 14.2 | [ 11.6; 17.5] |
| Classen 2011 | 354 | 795 | 44.5 | [ 40.1; 49.4] |
| Haukland 2017 | 253 | 812 | 31.2 | [ 27.5; 35.2] |
| Griffin 2008 | 138 | 854 | 16.2 | [ 13.7; 19.1] |
| Wilson 2012,Country G | 77 | 930 | 8.3 | [ 6.6; 10.4] |
| Wilson 2012,Country E | 76 | 931 | 8.2 | [ 6.5; 10.2] |
| Rutberg 2014 | 271 | 960 | 28.2 | [ 25.1; 31.8] |
| Wilson 2012,Country D | 146 | 984 | 14.8 | [ 12.6; 17.5] |
| Perez Zapata 2022 | 599 | 1,132 | 52.9 | [ 48.8; 57.3] |
| Wilson 2012,Country A | 81 | 1,358 | 6.0 | [ 4.8; 7.4] |
| Suarez 2014 | 424 | 1,440 | 29.4 | [ 26.8; 32.4] |
| Wilson 2012,Country H | 305 | 1,661 | 18.4 | [ 16.4; 20.5] |
| Mevik 2016 | 447 | 1,680 | 26.6 | [ 24.3; 29.2] |
| Wilson 2012,Country C | 281 | 1,938 | 14.5 | [ 12.9; 16.3] |
| Hommel 2020 | 2,116 | 1,998 | 105.9 | [101.5; 110.5] |
| Landrigan 2010 | 588 | 2,341 | 25.1 | [ 23.2; 27.2] |
| Nilsson 2016 | 658 | 3,301 | 19.9 | [ 18.5; 21.5] |
| Wilson 2012,Country B | 93 | 3,769 | 2.5 | [ 2.0; 3.0] |
| Wilson 2012,Country F | 218 | 3,977 | 5.5 | [ 4.8; 6.3] |
| Mayor 2017 | 430 | 4,833 | 8.9 | [ 8.1; 9.8] |
| Rutberg 2016 | 950 | 4,994 | 19.0 | [ 17.9; 20.3] |
| Kennerly 2014 | 3,430 | 9,017 | 38.0 | [ 36.8; 39.3] |
| Deilkas 2017,Norway | 1,672 | 10,986 | 15.2 | [ 14.5; 16.0] |
| Garrett 2013‡ | 6,573 | 17,295 | 38.0 | [ 37.1; 38.9] |
| Deilkas 2017,Sweden | 3,217 | 19,141 | 16.8 | [ 16.2; 17.4] |
| Nilsson 2018 | 8,104 | 56,447 | 14.4 | [ 14.0; 14.7] |
| **Random effects model** | **36,015** | **162,433** | **30.0** | **[ 23.9; 37.5]** |
| **Prediction interval** | | | | **[ 5.4; 164.7]** |

0   20   40   60   80   100   120   140

**Fig 3. Forest plot of adverse events per 100 admissions.** Ordered by sample size [5, 10, 15, 17–22, 34, 37–39, 45, 46, 50–54, 56–69, 71–79, 82–91, 93, 95–102]. In Wilson et al. 2012, countries were not further specified. (AEs, Adverse events; * pooled estimate; • mean estimate; ‡ calculated total number of AEs).

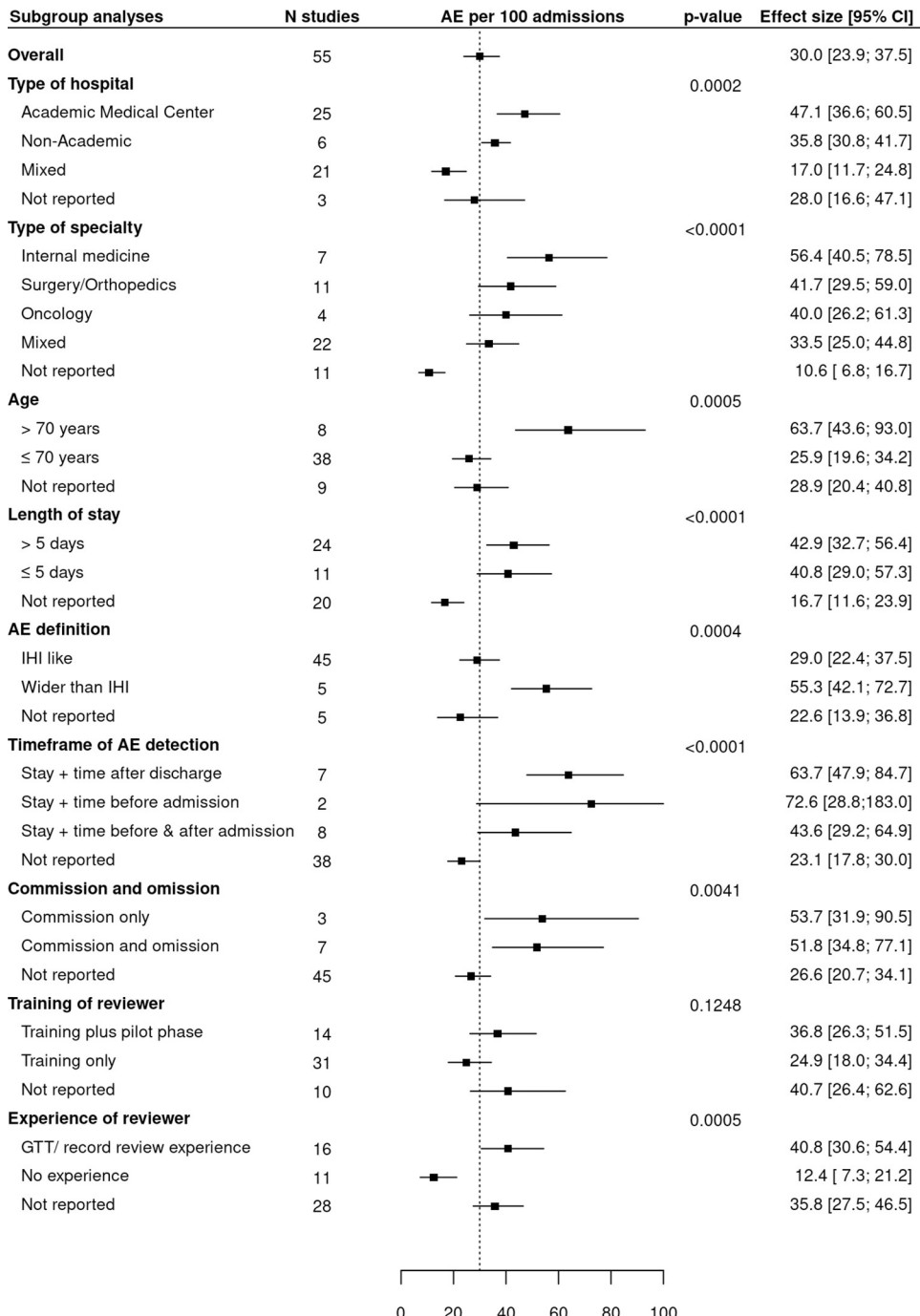

| Subgroup analyses | N studies | AE per 100 admissions | p-value | Effect size [95% CI] |
|---|---|---|---|---|
| **Overall** | 55 | | | 30.0 [23.9; 37.5] |
| **Type of hospital** | | | 0.0002 | |
| Academic Medical Center | 25 | | | 47.1 [36.6; 60.5] |
| Non-Academic | 6 | | | 35.8 [30.8; 41.7] |
| Mixed | 21 | | | 17.0 [11.7; 24.8] |
| Not reported | 3 | | | 28.0 [16.6; 47.1] |
| **Type of specialty** | | | <0.0001 | |
| Internal medicine | 7 | | | 56.4 [40.5; 78.5] |
| Surgery/Orthopedics | 11 | | | 41.7 [29.5; 59.0] |
| Oncology | 4 | | | 40.0 [26.2; 61.3] |
| Mixed | 22 | | | 33.5 [25.0; 44.8] |
| Not reported | 11 | | | 10.6 [ 6.8; 16.7] |
| **Age** | | | 0.0005 | |
| > 70 years | 8 | | | 63.7 [43.6; 93.0] |
| ≤ 70 years | 38 | | | 25.9 [19.6; 34.2] |
| Not reported | 9 | | | 28.9 [20.4; 40.8] |
| **Length of stay** | | | <0.0001 | |
| > 5 days | 24 | | | 42.9 [32.7; 56.4] |
| ≤ 5 days | 11 | | | 40.8 [29.0; 57.3] |
| Not reported | 20 | | | 16.7 [11.6; 23.9] |
| **AE definition** | | | 0.0004 | |
| IHI like | 45 | | | 29.0 [22.4; 37.5] |
| Wider than IHI | 5 | | | 55.3 [42.1; 72.7] |
| Not reported | 5 | | | 22.6 [13.9; 36.8] |
| **Timeframe of AE detection** | | | <0.0001 | |
| Stay + time after discharge | 7 | | | 63.7 [47.9; 84.7] |
| Stay + time before admission | 2 | | | 72.6 [28.8;183.0] |
| Stay + time before & after admission | 8 | | | 43.6 [29.2; 64.9] |
| Not reported | 38 | | | 23.1 [17.8; 30.0] |
| **Commission and omission** | | | 0.0041 | |
| Commission only | 3 | | | 53.7 [31.9; 90.5] |
| Commission and omission | 7 | | | 51.8 [34.8; 77.1] |
| Not reported | 45 | | | 26.6 [20.7; 34.1] |
| **Training of reviewer** | | | 0.1248 | |
| Training plus pilot phase | 14 | | | 36.8 [26.3; 51.5] |
| Training only | 31 | | | 24.9 [18.0; 34.4] |
| Not reported | 10 | | | 40.7 [26.4; 62.6] |
| **Experience of reviewer** | | | 0.0005 | |
| GTT/ record review experience | 16 | | | 40.8 [30.6; 54.4] |
| No experience | 11 | | | 12.4 [ 7.3; 21.2] |
| Not reported | 28 | | | 35.8 [27.5; 46.5] |

0   20   40   60   80   100

**Fig 4. Forest plot with stratified analysis of the nine selected study characteristics.** (AE, adverse event; CI, confidence interval; GTT, Global Trigger Tool; IHI, Institute for Healthcare Improvement; N Studies, number of studies).

the type of clinical specialty, for the category length of stay, the *not reported* category (n = 20, 36%) has a driving effect, with a mean of 16.7 AEs per 100 admissions (95% CI 11.6–23.9). Greater lengths of stay *(mean >5 days;* n = 24, 44%) had slightly higher AE rates than shorter ones (<*5 days;* n = 11, 20%) (respectively 42.9, 95% CI 32.7–56.4 and 40.8, 95% CI 29.0–57.3).

Almost all studies reported an *IHI-like definition* of AEs (n = 45, 82%). Of the five (9%) that *did not report* such a definition, AE rates were lower (respectively 29.0, 95% CI 22.4–37.5 and 22.6, 95% CI 13.9–36.8). The remaining five (9%) studies applying a *wider than IHI* AE definition reported clearly higher AE rates (55.3, 95% CI 42.1–72.7).

For the two characteristics, timeframe of AE detection and commission and omission the studies failed to report in 69% and 82% of the cases, seriously hampering the analyses. Studies that employed a *pilot phase* as part of the reviewer training (n = 14, 25%) might have had slightly higher detection rates than *training only* (respectively 36.8, 95% CI 26.3–51.5 and n = 31, 56%; 24.9, 95% CI 18.0–34.4). Reviewers with *no experience* in medical record review (n = 11, 20%) detected fewer AEs than those *with experience* (respectively 12.4, 95% CI 7.3–21.2) and n = 16, 29%; 40.9, 95% CI 30.6–54.4). Half of all studies *did not report* (n = 28, 51%) whether their reviewers had experience in medical record review. In those cases, the reported AE rates were comparable to those of experienced reviewers (35.8, 95% CI 27.5–46.5).

**Effect of risk of bias.**   Our quality assessment explained some of the variation regarding AE detection rates (S5 Fig). In eight studies (15%), patient selection was rated as *high* risk of bias because they included a slightly different patient population than defined in the inclusion criteria. These studies had higher rates of AEs than studies with a *low* risk of bias (respectively 61.2 vs. 32.5 AEs per 100 admissions). In studies where the risk of bias for the trigger tool methodology, the outcome category and the flow and timing were rated as *high* or *unclear*, considerably lower AE rates were detected than in those with a *low* risk of bias.

Similarly, regarding the trigger tool methodology's applicability-related concerns, ratings of *unclear* correlated with lower AE rates than those of *low* (respectively 10.7 vs. 38.7 AEs per 100 admissions).

## Discussion

The aim of this systematic review and meta-analysis was to synthesize AE detection rates with TT methodology and to explore variations in AE rates and assess the study quality in acute care inpatient settings. Reporting of study characteristics varied widely, and non-reporting of characteristics ranged from 5% to 82%. The summary estimate for AEs per 100 admissions was 30 (95% CI 23.9–37.5). An AE rate of 48 per 1,000 patient days, which translates into, 48 AEs in 200 patients with a length of stay of 5 days. Twenty-six percent of patients experience at least one AE related to their hospital stay and 63% out of all AEs were deemed preventable. Eight out of nine study characteristics explained variation in reported AE results. Studies conducted in academic medical centres, or with older populations reported higher AE rates than non-academic centres or younger adult populations. For several risk of bias categories (e.g., outcome, flow and timing), a higher risk of bias in a study indicated lower AE rates, which points to an underestimation of AE detection rates in low quality studies.

Analysing 17 studies in general inpatients, Hibbert et al. [3] reported AE rates of 8–51 per 100 admissions—a far smaller range than we detected (2.5–140). Our studies' larger range of AEs could result from our larger study sample (n = 54). Further, their rates of admissions with AEs ranged from 7% to 40%, with a cluster of nine falling between 20% and 29% [3]. We found a wider range—7%–69%, but the average (26%) is close to Hibbert et al. [3].

Schwendimann et al.'s scoping review [32] of multicentre studies reported a median of 10% of admissions with AEs, which is less than half what we found. But this is congruent with

Zanetti et al.'s integrative review, which reported between 5% and 11% [7]. Both of those reviews, especially Schwendimann et al.'s, concentrated solely on studies applying the HMPS methodology, not TT methodology [7, 32]. One possible reason for the lower rates could be that TT methodology requires the research team to include all identified AEs (if present, several AEs for one patient, not only the most severe, like in HMPS) [2, 12].

Interestingly, Panagioti et al.'s meta-analysis [6] found that half of their sample's AEs were preventable whereas our meta-analysis indicated an overall preventability of 61%. For an academic hospital with 32,000 annual admissions, a preventable percentage of 61 would mean roughly 5,000 AEs could be prevented annually–given effective prevention strategies could be implemented. The confidence intervals reported by Panagioti and our 95% CI largely overlaps despite the difference in selection criteria for inclusion. They included every study that explored AEs' preventability and many of those used the HMPS methodology, i.e., targeting more severe AEs [6].

Our meta-analysis explained part of the broad variation in AE detection via the selected study characteristics. One unanticipated finding was that, for many of these characteristics, essential details (e.g., length of stay) were not provided. For those, the *not reported* group had a dominant influence on AE detection rates. Although four study characteristics—type of specialty, length of stay, timeframe of AE detection, and commission and omission—showed differences in the subgroups, as the differences were driven by the *not reported* category, these only slightly explain the variation between AE detection rates. For all four characteristics, eight countries from which Wilson et al. [71] drew their samples fell within the *not reported* category, which might explain some of this result.

Compared to other categories, academic hospitals [34], higher patient age [75], and experienced reviewers [39] all corresponded with more AEs per 100 admissions. Supporting Sharek et al. [39] we found that experienced reviewers were less likely to miss AEs than unexperienced reviewers. These results support many published medical record review studies [23, 31–33]. Nevertheless, the findings need to be interpreted with some caution. Regarding type of specialty, the data on *internal medicine* and *surgery including orthopaedic* both involve wide confidence intervals (respectively 95% CI 40.5–78.5, and 95% CI 29.5–59.0); therefore, their higher numbers of AEs per 100 admissions (respectively 56.4 and 41.7) are to be questioned: numerous publications have found that surgical patients typically experience more AEs during their hospital stay than medical patients [6, 37, 103].

Addressing the overall quality of the included studies, we rated both their risk of bias and applicability-related concerns as *low*. This finding is supported by those of two earlier systematic reviews. First, Klein et al.'s [104] assessment of 24 of our 66 included publications indicated reasonable overall quality; second, also using a study sample that overlapped somewhat with ours, Panagioti et al. [6] rated all of the overlapping studies' risk of bias as low.

Nevertheless, regarding adherence to TT methodology, including data completeness and usability, our meta-analysis clearly showed that our overall study sample's reporting quality was inadequate. Our QAT explained part of the AE detection rate's high variability: where risk of bias is rated as *high* or *unclear* for "outcome", "trigger tool method" and "flow and timing", AE rates are lower than where risk of bias is rated as *low*. This suggests that insufficient reporting resulted in lower estimates, i.e., the actual AEs per 100 admissions are likely higher than reported here.

Although patterns of publication bias in the field of single arm studies measuring the incidence of AEs are not well understood, we decided to perform a funnel plot analysis to evaluate any association between small study size and the magnitude of the estimates of AEs per 100 admissions. Whenever an uncontrolled study evaluates effects and safety of a therapeutic intervention, publication bias may still be expected, where higher estimates of AE may be less likely

to be published. If this type of publication bias is associated with small study size, funnel plot exploration may detect it. The studies included in our review were more about health services and delivery research and we did not anticipate to find obvious signs of publication bias [105], which was eventually confirmed. The vast majority of studies did not report the occurrence of AEs per patient days. Rather than considering this as potential selective reporting bias, we reason that the field is insufficiently aware of the advantage of using person-time incidence rates over incidence proportions, where former facilitates comparison across studies.

## Strengths and limitations

Our systematic review was based on an exhaustive search strategy so that it is unlikely we missed studies that would have changed our findings. Throughout the search we have included two studies that were not identified with our search strategy. Those were lacking on of the core components like "adverse" [40] or "record" [86]. We did not do a systematic search of "grey literature" which may lead to remaining studies not identified.

In absence of a suitable risk of bias tool for the type of studies included, we adapted an existing QAT to simultaneously address risk of bias and applicability-related concerns of the included studies. We conducted stratified analyses not only to evaluate effects of studies' characteristics but also to evaluate effects of QAT domains. Our systematic review included a considerable high number of included studies when compared to previous reviews and resulted in a proportionately higher number of index admissions.

However, we also acknowledge further limitations. One was the exclusion of psychiatric, rehabilitation, emergency departments and intensive care settings. We set this criterion to maximize comparability across study settings. Similarly, by excluding studies focussed only on adverse drug events, we avoided skewing AE rates based on single-event results. Despite their benefits, both decisions reduced the final sample size.

Also, although we consider the identification and labelling of adverse events vital, we chose not to address either the types of AEs or their severity. Furthermore, we did not conduct an analysis of the influence of reported conflict of interest or funding in the included studies, which could further explain some of the variation. For the future, we also acknowledge that the registration of the review protocol on an open access repository is necessary.

Still, the most important limitation is that high levels of *not reported* information that hampered a full appreciation of the findings. The data did not allow to run multivariable models in a meaningful manner, so that all findings from univariable analyses need to be interpreted with caution, as we cannot exclude that some of the observed association, such as the effect of type of hospital, are confounded. For future studies on AEs via retrospective medical record review, irrespective of the detection methods used, the certainty of the evidence base would benefit from the standard use of a dedicated reporting guideline. Such a guideline is currently lacking for the type of studies included.

## Conclusion

Based on our analyses of 54 studies using TT methodology, we found an overall incidence of 30.0 AEs per 100 admissions—affecting 26% of patients. Of these we estimated that 63% were preventable, indicating a high potential to improve patient safety. However, lack of reporting and high levels of statistical heterogeneity limit these estimates' reliability.

Of nine TT study characteristics evaluated, our analyses indicate that eight explained part of the wide variation in AE incidence estimates. In four of those, most of the variation was driven by the not reported category (type of specialty, length of stay, timeframe of AE detection, commission and omission). For two characteristics (time frame of AE detection,

commission and omission), studies even failed to report the methodological information in 69% and 82%.

To enhance comparability—and the reporting of TT studies clearly needs improvement—we recommend the development and implementation of a reporting checklist accompanied with a guidance document specifically for studies on the use of retrospective medical record review methods for AE detection.

## Supporting information

**S1 Checklist. PRISMA 2020 checklist.**
(DOCX)

**S1 File. Quality assessment tool template.**
(PDF)

**S1 Table. Assessments of risk of bias and applicability-related concerns.**
(PDF)

**S1 Fig. Forest plot of AEs per 1000 patient days.** * = pooled estimate, • = mean estimate, ‡ = calculated total number of AEs, ~ = calculated total number of patient days [5, 15, 17–22, 34, 37, 39, 40, 45, 46, 50–52, 54, 57, 58, 60, 62–65, 67, 68, 72, 73, 76–79, 82, 84–87, 89–91, 93, 95, 96, 98–100, 102].
(TIF)

**S2 Fig. Forest plot percentage of admissions with at least one adverse event (AE).** CI, confidence interval; * = pooled estimate, • = mean estimate, + = calculated total number of admissions with ≥ 1 AE [5, 9, 14, 15, 17–22, 24, 34, 37, 39, 45, 46, 50–58, 60–68, 70, 72–87, 89–94, 96–101].
(TIF)

**S3 Fig. Forest plot percentage of preventable adverse events (AEs).** CI, confidence interval; * = pooled estimate, • = mean estimate, ¢ = calculated number of preventable AEs [15, 17–20, 34, 37–39, 46, 50, 51, 53, 59, 63–67, 71–75, 77, 78, 87, 89–91, 96–98, 100, 101].
(TIF)

**S4 Fig. Funnel plot for AEs per 100 admissions** [5, 10, 15, 17–22, 34, 37–39, 45, 46, 50–54, 56–69, 71–79, 82–91, 93, 95–102].
(TIF)

**S5 Fig. Forest plot with stratified analysis of the risk of bias and applicability-related concerns.** AE, adverse events; N studies, number of studies; CI, confidence interval [5, 10, 15, 17–22, 34, 37–39, 45, 46, 50–54, 56–69, 71–79, 82–91, 93, 95–102].
(TIF)

## Acknowledgments

The authors would like to thank Chris Shultis for the editing of this manuscript.

## Author Contributions

**Conceptualization:** Luisa C. Eggenschwiler, Anne W. S. Rutjes, Sarah N. Musy, Maria Unbeck, Michael Simon.

**Formal analysis:** Anne W. S. Rutjes.

**Investigation:** Dietmar Ausserhofer, Natascha M. Nielen, René Schwendimann, Maria Unbeck, Michael Simon.

**Methodology:** Luisa C. Eggenschwiler, Anne W. S. Rutjes, Sarah N. Musy, Maria Unbeck, Michael Simon.

**Project administration:** Sarah N. Musy, Natascha M. Nielen, Michael Simon.

**Resources:** Michael Simon.

**Supervision:** Sarah N. Musy, Maria Unbeck, Michael Simon.

**Validation:** Michael Simon.

**Visualization:** Luisa C. Eggenschwiler.

**Writing – original draft:** Luisa C. Eggenschwiler, Anne W. S. Rutjes, Sarah N. Musy, Dietmar Ausserhofer, Natascha M. Nielen, René Schwendimann, Maria Unbeck, Michael Simon.

**Writing – review & editing:** Luisa C. Eggenschwiler, Anne W. S. Rutjes, Sarah N. Musy, Dietmar Ausserhofer, Natascha M. Nielen, René Schwendimann, Maria Unbeck, Michael Simon.

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
