## [Decision Letter · Decision Letter 0]

13 May 2022

PONE-D-21-40420Variation in Detected Adverse Events using Trigger Tools: A Systematic Review and Meta-AnalysisPLOS ONE

Dear Dr. Simon,

Thank you for submitting your manuscript to PLOS ONE. After careful consideration, we feel that it has merit but does not fully meet PLOS ONE’s publication criteria as it currently stands. Therefore, we invite you to submit a revised version of the manuscript that addresses the points raised during the review process.

We look forward to receiving your revised manuscript.

Kind regards,

Mojtaba Vaismoradi

Academic Editor

PLOS ONE

Journal Requirements:

Reviewers' comments:

Reviewer's Responses to Questions

**Comments to the Author**

1. Is the manuscript technically sound, and do the data support the conclusions?

Reviewer #1: Partly

Reviewer #2: Yes

2. Has the statistical analysis been performed appropriately and rigorously? 

Reviewer #1: I Don't Know

Reviewer #2: Yes

3. Have the authors made all data underlying the findings in their manuscript fully available?

Reviewer #1: Yes

Reviewer #2: Yes

4. Is the manuscript presented in an intelligible fashion and written in standard English?

Reviewer #1: No

Reviewer #2: Yes

5. Review Comments to the Author

Reviewer #1: This is a systematic review and meta-analysis of 48 studies investigating the use of Trigger Tools for the assessment of adverse events in medical record review and estimating the rate of adverse events per 100 admission and several subgroups based on patient characteristics. The abstract does not adhere to PRISMA 2020 abstract, the method section does not adhere to PRISMA 2020 and the results section is difficult to follow as many results and analyses are reported. Furthermore, the last date of search is more than 12 months ago. To increase the readability and transparency of the reporting PRISMA 2020 should be followed and the result section revised.

Please see specific comments below.

Major:

1. The overall message of the study is difficult to follow, you report many results and subgroups(?) and these are not specified in the method section. Could you rearrange the result section with subheadings or omit some of the analyses to guide the reader.

2. The date of search is difficult to find, and it seems that the date of the last search >1,5-2 years ago. The search should be updated.

3. You state in the method section that PRISMA 2020 was identified but several items and the flow diagram from PRISMA 2020 is missing. I have listed some below in minor revisions but I recommend that you upload a PRISMA 2020 checklist stating where each item can be located.

Minor:

1. Acute care or acute-care? Please uniform

2. Incidence or prevalence

3. Abstract: please add the eligibility criteria on language and exclusion criteria that you describe in the method section.

4. Abstract: Please provide the date last searched (PRISMA 2020 for abstracts checklist. Item 4: https://www.equator-network.org/reporting-guidelines/prisma-abstracts/)

5. Abstract: Please describe methods to assess risk of bias (PRISMA 2020 for abstracts checklist. Item 5: https://www.equator-network.org/reporting-guidelines/prisma-abstracts/)

6. Abstract: Please report I^2

7. Abstract: I do not understand the results, could you simply? Several terms have not been introduced: e.g. applicability-related concerns, commission and omission, reviewers’ level of experience, the evidence on the remainder.

8. Abstract: Please provide details on registration and funding (PRISMA 2020 for abstracts checklist. Item 11+12: https://www.equator-network.org/reporting-guidelines/prisma-abstracts/)

9. Your REF 2 is the Trigger Tool – would ICH GCP not be better suited?

10. Consider using the term from the cited reference [8] “medical record review” rather that “record review” throughout your article.

11. Introduction: Please revise sentence and commas for: “Record review uses available data [8], was found to identify more AEs when 70 compared with many other methods [9, 10], can be repeated over time and can target specific AE 71 types, or the overall AE rate [11].”

12. Introduction: Please correct: “A "trigger" (or clue) consists either of specific wording or an event in a medical 82 record that could indicate the occurrence of an AE, e.g., readmissions within 30 days or pressure 83 ulcers [2].”

13. Methods: Please revise: “Design Systematic review and meta-analyses [27].” So that it reflect that you reported according to PRISMA 2020 [27].

14. Methods: Should the subheading “data sources” rather be “information sources” (PRISMA 2020, item 6)?

15. Methods: Your specific search strategy is difficult to follow:

a. You write that “The medical subject headings (MeSH) and keywords for titles and abstracts” was your search limited to title and abstract or were all fields searched (PRISMA 2020, item 7)?

b. Was “medical error” combined with AND or put in quotation marks?

c. Which of your search terms were Mesh terms? How were these translated from PubMed to the other databases?

d. Please provide the full (and specific) search strategy to each database as recommended by PRISMA 2020, item 7.

e. You first state that “Our search strategy was developed and validated using methods suggested by Hausner et 111 al. [28, 29]. This involves generating a test set, developing and validating a search strategy and 112 documenting the strategy using a standardized approach [29]” but later that “The detailed search strategy used 119 for this review is that described by Musy et al. [26].” – did you or reference [26] develop the search strategy and applied the methods.

16. Methods: Please provide date of last search (PRISMA 2020, 6), if the date of last search is >12 months ago I recommend that you update the search.

17. Methods: from you flow diagram, it seems that you have a restriction on the search date (2015 and onwards”, please report this, PRISMA 2020, item 6.

18. Methods: Were title and abstracts screened by one researcher or two researcher independently? Please specify in the manuscript.

19. Please add details on data collection process, PRISMA 2020, item 9.

20. Please add information on PRISMA 2020, item 10b.

21. Methods: why did you have to invent a new bias assessment tool?

22. Methods: How was heterogenicity measured, which cut-offs did you use?

23. Methods: your approach “Because R's 176 metaprop function does not accept proportions exceeding 100%, we adapted results of four 177 studies where the number of AEs exceeded the number of patient admissions. To reduce 178 oversized values to less than 100 AEs per 100 admissions, we reduced the number of AEs 179 detected to one less than the number of admissions (e.g., for a patient group of 240 with 336 180 AEs, we entered 239 AEs).” Can you provide a reference for this?

24. Results: dates should be reported in methods.

25. Results: Please help me understand your flow diagram – the layout of PRISMA 2020 has not been used. Did you use automatic tolls for the screening and exclusion of the 4531 non-trigger tools? Please specify in the method section if you did and only screened 406 title/abstracts independently. Only full-text exclusion reasons must be explained in detail.

26. Results: Please correct: “The reviewed studies were all published between 2009 and 2020” to “included”.

27. Results: please uniform: “Overall, we included 192,316 index admissions in our report” in the abstract these as described as patients, which is more correct?

28. Results: which type of studies was included in the review? Cohort studies, RCTs? This is not described in the method section or table 1.

29. Result: There are a lot of results reported in this section – and a lot of analysis. The section is difficult to follow and not all subgroups are evident from the method section. Could you omit so analysis or add some aiding subheadings?

30. Please add PRISMA 2020, item 22.

31. Discussion: Please provide a key findings paragraph in the beginning of the discussion section with the key findings of your study without references to other studies.

32. Discussion: please expand your limitations section.

33. Did you analyse conflict of interest and funding of included studies and accounted for that in the analyses?

Reviewer #2: TITLE

The title is clear with enough detail for the reader to know what to expect.

RELEVANCE AND ORIGINALITY

Adverse events are an ongoing occurrence in the health landscape and the mechanism of identifying and reporting adverse events is not consistent across or between countries. This review is relevant as it provides an argument (using a recognised high quality and rigorous approach, i.e., a systematic review and meta-analysis) for the need to address this inconsistency with clearer reporting guidance.

AUTHENTICITY AND REFERENCING

The manuscript appears to be the work of the author with appropriate attribution to the work of others both in text and in the reference list.

ABSTRACT/INTRODUCTION

The abstract is comprehensive and an accurate reflection of the manuscript. The introduction is brief yet provides enough information from the literature to support the need for the review. In addition, key terms, (e.g., ‘global trigger tool’, ‘trigger’) are explained and operationalised for the review. The introduction leads logically to the gap in the literature and the aims of the study.

AIMS

Dual aims are clear.

METHODOLOGY

The methodology is well described and replicable, apart from a few queries:

• One evidence source searched, ‘all authors’ personal libraries’ (line 117), is not defined or described. Are the authors referring to self -authored publications or simply publications amassed in personal collections? If the former, then these papers would presumably be indexed in one of the other databases searched. If the latter, then it renders the search not replicable. Removal of ‘personal libraries’ or explanation for its inclusion might address any concerns raised by its inclusion.

• Similarly, an explanation for the choice of the three journals hand searched would allay any concerns of bias in the search strategy.

Re eligibility criteria – point #3 is “…acute care (including elective admissions) hospital settings… (line 122). I would think elective admissions are inherently a part of an acute care cohort. Did the authors mean ‘emergency admissions’? In either case, this could be clarified.

Table 1 is particularly helpful.

RESULTS

Results are comprehensive and well organised.

TABLES AND FIGURES

Tables and figures are well presented and do not replicate information in text.

DISCUSSION/CONCLUSION

The discussion is supported by the findings and the findings are situated within the current body of evidence on the topic. Recommendations for future practice related to adverse events and future research reporting on adverse events, albeit very brief (e.g., one sentence each), logically derives from the findings and discussion.

OTHER COMMENTS

Use of a reporting guideline is not evident but is a conventional expectation. The authors might consider adding reference to this in some way.

WRITING STYLE

The writing style is easy to academically sound and easy to read.

SCHOLARLY APPROACH

The authors have used a scholarly approach that begins with a clearly stated premise so that compelling arguments can be presented and supported with up-to-date literature, including empirical research evidence. Providing more critique of the studies cited in the introduction and in the discussion would elevate this further.

OVERALL COMMENTS

My comments have been provided in the spirit of collegiality to hopefully assist the authors in further preparing their manuscript for publication. I commend the authors on this high-quality report of their systematic review and meta-analysis.

6. PLOS authors have the option to publish the peer review history of their article (what does this mean?). If published, this will include your full peer review and any attached files.

Reviewer #1: **Yes: **Siv Fonnes

Reviewer #2: **Yes: **Sonya R Osborne

---

## [Author Response · Author response to Decision Letter 0]

24 Jun 2022

We appreciate the opportunity to address the very helpful reviewers’ comments and revise our manuscript. Below, please find item-by-item responses to the Reviewers’ comments, which are included verbatim. All page and paragraph numbers refer to locations in the revised manuscript.

---

## [Decision Letter · Decision Letter 1]

16 Aug 2022

Variation in Detected Adverse Events using Trigger Tools: A Systematic Review and Meta-Analysis

PONE-D-21-40420R1

Dear Dr. Simon,

We’re pleased to inform you that your manuscript has been judged scientifically suitable for publication and will be formally accepted for publication once it meets all outstanding technical requirements.

Kind regards,

Prof, Mojtaba Vaismoradi, PhD, MScN, BScN

Academic Editor

PLOS ONE

Additional Editor Comments (optional):

Reviewers' comments:

Reviewer's Responses to Questions

**Comments to the Author**

1. If the authors have adequately addressed your comments raised in a previous round of review and you feel that this manuscript is now acceptable for publication, you may indicate that here to bypass the “Comments to the Author” section, enter your conflict of interest statement in the “Confidential to Editor” section, and submit your "Accept" recommendation.

Reviewer #1: All comments have been addressed

Reviewer #2: All comments have been addressed

2. Is the manuscript technically sound, and do the data support the conclusions?

Reviewer #1: Yes

Reviewer #2: Yes

3. Has the statistical analysis been performed appropriately and rigorously? 

Reviewer #1: Yes

Reviewer #2: Yes

4. Have the authors made all data underlying the findings in their manuscript fully available?

Reviewer #1: Yes

Reviewer #2: Yes

5. Is the manuscript presented in an intelligible fashion and written in standard English?

Reviewer #1: Yes

Reviewer #2: Yes

6. Review Comments to the Author

Reviewer #1: Thank you for your comprehensive work on revising and improving your systematic review and meta-analysis. The reporting according to PRISMA 2020 has improved the readability and transparency of the manuscript. The revision of the results section and the key findings paragraph in the discussion section has made the message and the results of your study easier to understand and follow. Congratulations on your comprehensive and hard work.

Reviewer #2: The authors have addressed all of my comments in the revision. I acknowledge the data has been updated in light of an updated search.

---

## [Editor Report · Acceptance letter]

23 Aug 2022

PONE-D-21-40420R1 

Variation in detected adverse events using trigger tools: A systematic review and meta-analysis 

Dear Dr. Simon:

I'm pleased to inform you that your manuscript has been deemed suitable for publication in PLOS ONE. Congratulations! Your manuscript is now with our production department. 

Kind regards, 

on behalf of

Professor Mojtaba Vaismoradi 

Academic Editor

PLOS ONE